# Relationship between the Cycle Threshold Value (Ct) of a *Salmonella* spp. qPCR Performed on Feces and Clinical Signs and Outcome in Horses

**DOI:** 10.3390/microorganisms11081950

**Published:** 2023-07-30

**Authors:** Hélène Amory, Carla Cesarini, Lorie De Maré, Clémence Loublier, Nassim Moula, Johann Detilleux, Marc Saulmont, Mutien-Marie Garigliany, Laureline Lecoq

**Affiliations:** 1Equine Clinical Department, Faculty of Veterinary Medicine, Bât. B41, University of Liège, Sart Tilman, 4000 Liège, Belgium; ccesarini@uliege.be (C.C.); lorie.demare@uliege.be (L.D.M.); cloublier@uliege.be (C.L.); jdetilleux@uliege.be (J.D.);; 2Fundamental and Applied Research for Animals & Health (FARAH), Faculty of Veterinary Medicine, University of Liège, Sart Tilman, 4000 Liège, Belgium; nassim.moula@uliege.be (N.M.); mmgarigliany@uliege.be (M.-M.G.); 3Department of Veterinary Management of Animal Resources, Faculty of Veterinary Medicine, Bât. B41, University of Liège, Sart Tilman, 4000 Liège, Belgium; 4Regional Animal Health and Identification Association (ARSIA), 2 Allée des Artisans, ZA du Biron, 5590 Ciney, Belgium; 5Department of Morphology and Pathology, Faculty of Veterinary Medicine, Bât. B41, University of Liège, Sart Tilman, 4000 Liège, Belgium

**Keywords:** SIRS, equine, salmonellosis, fecal shedding, bacterial load, qPCR

## Abstract

The objective of this retrospective study was to evaluate the clinical significance of fecal quantitative real-time polymerase chain reaction (qPCR) *Salmonella* results when taking the cycle threshold values (Ct) into account. The study included 120 *Salmonella* qPCR-positive fecal samples obtained from 88 hospitalized horses over a 2-year period. The mean Ct of the qPCR test was evaluated in regard to (1) clinical outcome and (2) systemic inflammatory response syndrome (SIRS) status (no SIRS, moderate SIRS, or severe SIRS) of the sampled horses. An ROC analysis was performed to establish the optimal cut-off Ct values associated with severe SIRS. The mean ± SD Ct value was significantly lower in samples (1) from horses with a fatal issue (27.87 ± 5.15 cycles) than in surviving horses (31.75 ± 3.60 cycles), and (2) from horses with severe SIRS (27.87 ± 2.78 cycles) than from horses with no (32.51 ± 3.59 cycles) or moderate (31.54 ± 3.02 cycles) SIRS. In the ROC analysis, the optimal cut-off value of Ct associated with a severe SIRS was 30.40 cycles, with an AUC value of 0.84 [95% confidence interval 0.76–0.91] and an OR of 0.64 [0.51–0.79]. Results suggest that including the Ct value in the interpretation of fecal qPCR results could improve the diagnostic value of this test for clinical salmonellosis in horses.

## 1. Introduction

In human and veterinary medicine, acute diarrhea is associated with significant morbidity and mortality [1,2,3]. According to the World Health Organisation (WHO), there are around 2000 billion cases of acute diarrhea in humans diagnosed per year worldwide, among which around one third are considered to be foodborne [2,4]. *Salmonella* spp. infections are considered to be one of the most frequent causes of diarrhea [2,5]. Globally, in 2010, the WHO estimated the number of salmonellosis cases worldwide at 157 million, of which 56,969 had a fatal outcome and almost half were foodborne [6]. In the USA, the Center for Disease Control and Prevention (CDC) estimates that, each year, Salmonella infection is responsible for approximately 1.35 million illnesses, 26,500 hospitalizations, and 420 deaths [3]. According to the European Centre for Diseases Prevention and Control (ECDC), salmonellosis is the second most frequently reported cause of gastrointestinal infection in Europe, with 60,494 confirmed cases in 2021, 73 of which had a fatal outcome [7]. Salmonellosis is therefore considered to be a zoonosis of major importance [2,3,7].

The animal species most frequently incriminated as a source of infection in foodstuffs are poultry and swine, mainly through eggs and meat [2,7,8]. However, other animal species, including horses, can also be involved from time to time [8,9]. Another important source of infection from animal origin is contact with sick animals, and direct or indirect contact with sick horses is considered to be a significant zoonotic risk factor for the transmission of *Salmonella* spp. [4,10,11,12], especially as the strains involved in nosocomial equine infections are often multi-drug resistant (MDR) [13,14]. In a recent study carried out over 1 year in an equine hospital, 22 different strains of *Salmonella* were isolated from 3–17% of the samples obtained in areas where the staff had exclusive access, which means that the horses did not have direct contact with those areas. From the 22 strains isolated, 19 were resistant to at least one of the antibiotics tested and 9 were MDR [15].

*Salmonella* are Gram-negative, facultative, rod-shaped bacteria belonging to the Enterobacteriaceae family. The *Salmonella* genus comprises two species, *Salmonella bongori* and *Salmonella enterica*, of which *S*. *enterica* is the type species [5]. Based on genome analysis and biochemical characteristics, six subspecies of *S. enterica* have been identified: *enterica*, *salamae*, *arizonae*, *diarizonae*, *houtenae* and *indica*. On the basis of antigenic characteristics, the subspecies have been subdivided into serogroups (O antigen) and serovars (H antigen). To date, more than 2500 serovars have been identified [2,5,16,17,18]. All species and subspecies are considered potentially pathogenic, but the virulence of many of them is unknown. The vast majority of clinical cases, however, are associated with the subspecies *S. enterica* I (usually referred to as *S. enterica* subsp. *enterica*), which comprises around 1450 serovars, responsible for 99.5% of mammalian infections [18].

*Salmonella enterica* serotype *Enteritidis*, *Typhimurium* and monophasic *S. typhimurium* are the three most important serotypes of *Salmonella* transmitted from animals to humans in most parts of the world. These serovars accounted for almost 80% of the cases of human salmonellosis in the European Union over the last years. *S. infantis* has been consistently the fourth most frequently reported serovar in domestically acquired and travel-associated infections. Since 2018, *S. newport* became the fifth most common strain isolated, replacing *S. derby* [19].

In equids, the serovars involved in clinical cases of salmonellosis are numerous and vary according to time, geographical location, and outbreaks [5,20]. They have mainly been documented in the USA, with the most frequently reported serovars being *S. thyphimurium*, *S. agona*, *S. newport*, *S. anatum,* and *S. braenderup* [5,21,22]. The serovars circulating in Europe have been less documented, but published data for the Netherlands and the UK report that *Salmonella thyphimurium* appears to be the most common strain isolated also. *Salmonella enteritidis*, *S. anatum,* and *S. agama* are also regularly incriminated [20,23,24,25]. 

*Salmonella* infections in humans are essentially foodborne, which leads to a close monitoring of food-producing animals (especially swine and poultry); conversely, serovars circulating in companion animals, including horses, have only been monitored to a limited extent. However, prevalence and resistance profile studies carried out in those species highlight the importance of monitoring them also [26,27]. The detection of MDR strains of *Salmonella* seems to be increasing in horses, especially in hospitalized animals [26,27,28]. In the USA, a long-term study in foals showed an increase in the resistance of isolated strains of *Salmonella* to ceftizoxime (a third-generation cephalosporin) and a decrease in their sensitivity to gentamicin [29]. The extensive spread of the serovar *Typhimurium* MDR phage type DT104 has been reported in animals and humans since the 1990s [30], and more recently an extension of the fluoroquinolone (ciprofloxacin)-resistant serovar *Kentucky* ST198 has also been documented in dogs, horses, and humans [31].

*Salmonella* spp. is the most frequent bacteria associated with nosocomial infections in the equine species and is a major cause of diarrhea and systemic inflammatory response syndrome (SIRS) in adult horses [5,15,32,33,34]. *Salmonella* infection increases the risk of mortality, can be a source of considerable additional expenses for treatment and biosecurity measures, and might even lead to the temporary closure of veterinary hospitals [15]. Considering this and all the aforementioned zoonotic risks, state of the art practice in equine hospitals therefore requires active surveillance of this pathogen by controlling its fecal shedding in the hospitalized horses and in the environment [12,14,35,36].

Fecal culture is considered the gold standard technique for diagnostic confirmation of *Salmonella* shedding. It requires a strictly controlled methodology, including specific enrichment techniques and repeated sampling (3 to 5, taken 12–24 h apart) because shedding is often intermittent [28,37]. However, this technique is slow, with the culture requiring at least 48 h and the enrichment steps further prolonging the detection time, which renders it poorly efficient for early detection of shedders and prevents rapid implementation of biosecurity measures when needed [38,39]. Moreover, the bacterial culture often lacks sensitivity, with the growth of the bacteria being affected by sampling and handling methodology, culture technique, prior treatment with anti-microbials, or fecal bacterial load [40]. Therefore, the risk of false negative results is relatively high with the culture. Moreover, interpretation of a positive result on fecal samples obtained from hospitalized animals is not easy because asymptomatic *Salmonella* shedding is not uncommon in horses [5,37]. The use of PCR techniques brings the advantages of being faster and more sensitive than culture techniques, and therefore can be used as the first line to detect shedders [39]. Unfortunately, a positive PCR result indicates the presence of the bacteria’s genetic material but precludes from determining its pathogenic potential, hence having a high risk of false positive results with this technique. PCR also precludes from evaluating the phenotypic characteristics of the strain [40,41,42]. When a positive result is obtained by PCR, it is therefore recommended to submit several fecal samples to culture [5,39], but in the meantime, before obtaining the results of the culture, biosecurity measures must be implemented, which is restrictive, costly, and sometimes poorly accepted by the nursing staff or the owner, particularly in cases that will not subsequently develop overt clinical signs.

Quantitative real-time PCR (qPCR) provides quantitative results, and the cycle threshold value (Ct, number of cycles required for the PCR to detect the target microorganism) can be used as an indirect estimate of the load of the pathogen in the sample, as it has already been demonstrated in the detection of *Salmonella enteritidis* in feces of poultry [43]. In nasopharyngeal swabs of humans infected with SARS-CoV-2, the Ct values were correlated with viral loads and with the severity of the clinical signs, which further increased the clinical interest of this test [44]. Such a correlation has never been studied for salmonellosis, but could be useful to refine the rapid interpretation of the qPCR results. Indeed, it might aid in differentiating between clinical and sub-clinical shedders and in better assessing the risks of nosocomial infections in equine patients as it has been previously suggested but never tested in horses [38].

The objective of this study was to evaluate whether the Ct value of a diagnostic qPCR test for *Salmonella* spp. using a broad spectrum primer has a predictive value for clinical disease in the horses in our study population. The hypothesis was that positive qPCR results with lower Ct values would be more often associated with clinical signs of severe disease, as evaluated by the SIRS score.

## 2. Materials and Methods

### 2.1. Study Population

This retrospective study was conducted on cases admitted to the equine teaching hospital of the University of Liege from December 2019 to December 2021, and that tested positive for PCR for *Salmonella* in the feces. The horses were tested because they were presented to the clinic for gastrointestinal problems or as part of the hospital biosecurity protocol if they developed fever and leucopenia of unknown origin with or without diarrhea during hospitalization. 

### 2.2. Sampling Procedures

Each case had one to five fecal samples taken during hospitalization. Samples were taken by rectal palpation, kept at 4 °C until analysis, and submitted to the laboratory for *Salmonella* spp. qPCR assay (see below) within 48 h. 

For each case that tested positive for *Salmonella* spp. on qPCR, specific patient’s data at the time of sampling of the feces were retrospectively collected from the clinical file. These data included age, gender, and breed of the animal, the clinical issue (dead or euthanized during hospitalization or discharged from the hospital), and clinical and clinicopathological parameters enabling an SIRS score to be established. The SIRS score used was established on a scale of 0 to 4 based on the number of criteria present amongst: (1) heart rate > 56 bpm, (2) respiratory rate > 20 rpm, (3) rectal temperature < 37 °C or > 38.5 °C, and (4) circulating white blood cell count (WBC) < 5 × 109/L or >12.5 × 109/L [45]. When no WBC data were available at the time of fecal sampling, these criteria were considered to be absent in determining the SIRS score. For the purpose of this study, horses were classified into 3 groups according to this score: non-SIRS (SIRS score 0 or 1), moderate SIRS (SIRS score 2), and severe SIRS (SIRS score 3 or 4). Additionally, the final outcome (i.e., discharged from the hospital or dead/euthanized) and the presence or not of clinical signs suggestive of salmonellosis during the whole hospitalization were recorded for each case. This was defined as the presence of at least 2 out of the 3 following criteria: diarrhea (more than 2 loose or liquid feces over 24 h), fever (rectal temperature > 38.5 °C), and leucopenia (white blood cell count < 5 × 109/L) [14,46].

### 2.3. Quantitative Real-Time PCR Assay

Two hundred milligrams of feces were subjected to DNA extraction using the Nucleospin^®^ DNA Stool Kit, as described by the manufacturer (Macherey-Nagel GmbH & Co., Kg, Düren, Germany). Quantitative qPCR for the amplification of the *Stn* target gene DNA was performed on a QuantStudio1 thermocycler (ThermoFisher Scientific, Brussels, Belgium), according to Moore and Feist (2007) [47]. Briefly, qPCR reactions (20 µL) included 2 µL of DNA, 10 µL of Luna Universal Probe qPCR Master Mix (Bioké, Leiden, The Netherlands), 6 µL of water, and 2 µL of the primers mix. The primer mix contained 20 µL of a 10 µM solution of each primer (F: 5′-GCCATGCTGTTCGATGAT; R: 5′-GTTACCGATAGCGGGA) and 10 µL of a 10 µM solution of the probe (FAM-5′-TTTTGCACCACMGCCAGCCC-3′-IBFQ). PCR conditions were as follows: 5 min at 95 °C, followed by 45 cycles at 95 °C for 30 s, 55 °C for 30 s, and 72 °C for 1 min; then a final extension at 72 °C for 3 min. All qPCR samples were analyzed in duplicates. A bovine isolate of *Salmonella enterica* Serovar *Dublin* kindly provided by the Association Régionale de Santé et d’Identification Animales (ARSIA, Ciney, Belgium) was used as a positive control. β-Actin was used as an extraction and qPCR control, according to Moore and Feist (2007) [48], with the same conditions as for the *Salmonella* spp. qPCR.

To evaluate the correlation between the qPCR Ct value and the actual bacterial concentration, three Salmonella strains, isolated from horse samples at the ARSIA (Ciney, Belgium, and identified by seroagglutination on the slide, according to the Kauffmann White protocol, were used to establish standard curves, using the same extraction and qPCR methods as detailed above. One strain corresponded to *Salmonella enterica* ssp. *enterica* serovar *Coeln* and two strains to the *Enteritidis* serovar. 

### 2.4. Statistical Analysis

The mean qPCR Ct values and the log CFU/mL were compared using an unpaired *T*-test between samples obtained from cases that survived or had a fatal outcome.

The mean qPCR Ct values and the log CFU/mL of the samples obtained from the different SIRS groups were compared using a one-way ANOVA test.

A receiver-operating characteristic (ROC) analysis was conducted to compute the sensitivity and specificity of the fecal *Salmonella* qPCR Ct value as an indicator of the simultaneous development of a severe SIRS.

Statistical significance was established at *p*-values less than 0.05.

## 3. Results

During the 2-year study period, 399 equine fecal samples were collected from 287 cases and submitted for *Salmonella* qPCR testing. Amongst them, 120 samples tested positive and were obtained from 82 horses of various breeds. The mean ± SD age of these horses was 11.2 ± 8.2 years (median = 11.0; range 1 day–29 years) with 33 (40%) mares, 36 (44%) geldings, and 13 (16%) stallions (Appendix A ). Amongst those horses, 57 (70%) presented clinical signs suggestive of salmonellosis (i.e., at least two out of the three following criteria: fever, diarrhea, and leucopenia) [14,46] during hospitalization.

Interestingly, the standard curves established using three horse isolates of Salmonella, corresponding to two serovars (*S. enterica* ssp. *Enterica* serovar *Coeln* (1 isolate) and *S. enterica* ssp. *Enterica* serovar *Enteritidis* (2 isolates)), showed an excellent correlation between the qPCR Ct value and the bacterial concentration expressed as log CFU/mL (R^2^ values of 0.9983, 0.9973, and 0.9997, respectively; see Appendix A). In addition, the estimated bacterial concentration based on the qPCR Ct value was relatively close using the three different equations obtained (standard deviation of 15.9% for a Ct value of 30, for instance; Appendix A). Each Ct value was then converted to “predicted” log CFU/mL by averaging the three equations obtained for the reference strains used.

The mean ± SD Ct value from the 120 positive samples was 31.45 ± 3.74 cycles (range 19.9 to 40.8; 95% CI 30.8 to 32.1), which, after conversion into log CFU/mL, corresponded to mean ± SD values of 3.83 ± 1.11 log CFU/mL (range 1.04 to 7.26; 95% CI 3.62 to 4.02). Amongst these samples, 72 were obtained from horses presenting no SIRS, 26 from horses presenting a moderate SIRS, and 22 from horses presenting a severe SIRS at the time of sampling. The mean ± SD Ct values were statistically different between the samples obtained from horses presenting a different SIRS category, the Ct values obtained from horses presenting severe SIRS (27.87 ± 2.78 cycles) [95% confidence interval 26.6–29.7] (after conversion into log CFU/mL: 4.89 ± 0.83 log CFU/mL [95% confidence interval 4.52–5.26]) being significantly lower (*p* < 0.001) than the Ct values obtained from horses presenting no SIRS (32.51 ± 3.59) [95% confidence interval 31.7 to 33.4] (after conversion into log CFU/mL: 3.51 ± 1.07 log CFU/mL [95% confidence interval 3.25–3.76]) or a moderate SIRS (31.54 ± 3.02) [95% confidence interval 30.3–32.8] (after conversion into log CFU/mL: 3.80 ± 0.90 log CFU/mL [95% confidence interval 3.43–4.15]) at the sampling time (Figure 1).

The ROC analysis gave an optimal cut-off Ct value of 30.40 cycles for association with signs of severe SIRS (as compared with no or moderate SIRS signs) at the sampling time, with an area under curve of 0.84 [95% confidence interval 0.76–0.91] and an OR of 0.64 [0.51–0.79] (Figure 2).

In total, 9 of the 82 (11%) cases included in the study died or were euthanized. The mean ± SD Ct value of the samples obtained from those horses (28.74 ± 4.99 cycles) [95% confidence interval 24.9–32.6] was significantly lower (*p* = 0.022) (after conversion into log CFU/mL: 4.63 ± 1.49 log CFU/mL [95% confidence interval 3.49–5.77]) than the mean ± SD Ct of the samples obtained from the cases that were discharged from the hospital (31.75 ± 3.61 cycles) [95% confidence interval 31.1–32.4] (after conversion into log CFU/mL: 3.73 ± 1.07 log CFU/mL [95% confidence interval 3.53–3.94]) (Figure 3). 

## 4. Discussion

In this study, fecal samples from hospitalized horses considered at high risk for shedding of *Salmonella* spp. were tested using a qPCR targeting the *Stn* gene, and a high percentage of the samples (120/399 = 30.1%) were positive. Results suggest a link between the Ct value of the qPCR and the severity of the clinical signs of salmonellosis. Mean Ct values were significantly lower in horses with a fatal outcome compared to survivors, and Ct values of the samples from horses showing signs of severe SIRS were significantly lower compared with those obtained when horses were showing either no or moderate signs of SIRS.

Twenty-nine percent (29%) of the horses included in this study tested positive for *Salmonella* at least once using a qPCR targeting the *Stn* gene. This percentage is difficult to compare with *Salmonella* shedding rates reported in previous studies, because selection criteria, diagnostic techniques, genes targeted, primers for the qPCR, and the numbers of samples per case vary greatly among studies [36]. 

Regarding the cases selection, *Salmonella* carriage is known to be influenced by numerous risk factors in the horse, the most commonly mentioned being hospitalization for a gastrointestinal problem (colic or diarrhea) or a major illness with systemic repercussions [34,36,49,50], especially with prolonged hospitalization [51,52]. Other risk factors of *Salmonella* shedding in horses have been identified, some of which are still controversial or have not been statistically tested [33,34,36]. Among these, the age of the horse, administration of antibiotics or anthelmintics, surgery, general anesthesia, transport, diet changes, unsanitary conditions, or overcrowding have been mentioned [5,22,28,33,34,36].

Even in studies using PCR and targeting samples from horses with gastrointestinal problems, the reported *Salmonella* shedding rates vary greatly, between 2 to 68% [39,46,52,53,54,55,56]. The rate of positive samples obtained in this study is rather high compared with most of the previous studies using PCR. This could partly be explained by the fact that most of the horses tested (70%) presented signs suggestive of salmonellosis during hospitalization. However, it could also be linked to the sensitivity of the PCR primer used in the present study, targeting the *Stn* gene rather than the *SpaQ* [21,42,57], *HisJ* [40,41,53], or *InvA* [38,39] genes as previously reported in equine studies. In various species, the use of *InvA* genes has long been regarded as the most reliable in *Salmonella* discrimination since many strains possess the *InvA* gene within their genomes. The presence of this virulence factor has been linked to the ability of *Salmonella* to invade the epithelial tissues of the host [58]. However, *Salmonella* associated infection of the intestinal mucosa and translocation through the intestinal barrier during infection is not only mediated by invasion genes; other genes have been demonstrated to be required for bacterial colonization of the host intestine [59], and several studies have shown that a significant number of *Salmonella* strains do not possess the *InvA* gene [47,58,60,61].

The *Salmonella Stn* gene is an interesting target for PCR, because it is one of the few reported genes that has been able to consistently correctly identify all *S. enterica* with no false positives [61,62,63,64,65,66]. The study of Moore and Feist (2006) [47] developed a *Salmonella*-specific qPCR targeting *Stn* sequences conserved in both *Salmonella* species (*S. enterica* and *S. bongori*) and all serotypes, and demonstrated that *Stn* is a valid target for both species of *Salmonella*, having 100% inclusivity for the 269 *Salmonella* isolates tested, including 255 *Salmonella enterica* representing 158 serotypes, and for 14 *Salmonella bongori* representing 12 serotypes, 96.4 exclusivity for the 84 non-*Salmonella* isolates tested representing 56 species from 31 genera, and a level of detection of 3 CFU for cultured *Salmonella* spp. In a study investigating the antibacterial activity of an aqueous extract of *Ilex paraguariensis* against 32 different strains (*S. enteridis*, *S. typhimurium* and *S. corvalis*) of nontyphoidal *Salmonella* isolated from human patients with clinical disease, the *Stn* gene was detected in 100% of the *Salmonella* spp. isolates, while the *SdiA* gene was detected in 81.2% of the isolates, and *InvH*, *SopE*, *InvA,* and *SopB* genes were only detected in 37.5, 37.5, 28.1, and 28.1% of the isolates, respectively [61]. This study also suggests a low sensitivity for the commonly used qPCR targeting the *invA* gene in detecting pathogenic *Salmonella* strains. The advantage of using a broad-spectrum primer such as *Stn* is that it increases the sensitivity rate of *Salmonella* carriage detection and reduces the risk of transmission of the bacteria in hospitalized patients. The nature and severity of the clinical signs associated with fecal shedding of *Salmonella* in horses vary greatly, ranging from asymptomatic carriage to severe septic shock, often associated with fever, diarrhea or reflux, and/or leucopenia, which can lead to death in the most severe cases [5,37,52]. In horses hospitalized for colic, *Salmonella* carriage has been associated with more severe pathologies or cases requiring surgery [2], and with an increased rate of complications and mortality compared to non-carriers during hospitalization [46] or after hospital discharge [67]. Studies on the impact of *Salmonella* shedding on patient’s outcome are, however, controversial, with some studies not having demonstrated an increased risk of mortality in carrier horses during hospitalization [68] or upon return to the stable [46]. However, it is worth noting that none of these studies assessed shedding quantitatively. The survival prognosis could perhaps be improved based on such an approach.

The clinical repercussions of intestinal colonization by *Salmonella* in the horse de-pend on the interaction between, on one hand, the state of health and immunity of the host (e.g., stress, immune status, concurrent gastrointestinal disease), and, on the other hand, the infective dose and the virulence of the *Salmonella* strain involved [5,22]. Bacterial load is certainly not the only factor that conditions the severity of clinical signs when they develop. Nevertheless, the results of the present study highlight, interestingly, an association between low Ct values of qPCR detection of *Salmonella* on fecal samples and the severity of simultaneous clinical signs. Since the Ct value is known to be inversely proportional to the amount of target DNA in a sample and can therefore be used as a rough indirect estimate of its infectious load [44], the hypothesis of a higher fecal *Salmonella* shedding in more severely affected horses is plausible. The Ct value of qPCR has been shown to be related to the *Salmonella* load in various kinds of samples. It has been used to determine the detection threshold for viable but non-culturable *Salmonella typhimurium* in sewage sludge [69], or for *Salmonella enteritidis* detection in pooled environmental samples from poultry production units [43]. In a study comparing culture and qPCR results for the detection of *Salmonella* spp. in the feces of hospitalized horses [38], the number of copies of the *InvA* gene (a target gene often used in Salmonella PCR tests in horse) detected by qPCR was significantly higher, suggesting a higher bacterial load in the samples that were culture positive than in those that were culture negative. In the conclusions of this paper, the authors suggested future studies to establish how quantitative PCR-based assays could be used to differentiate between clinical and sub-clinical shedders and to assess the risk of infection to other hospitalized animals. Similarly, we observed a strong correlation between the *Stn* qPCR Ct value and the bacterial concentration (CFU/mL), based on serial dilutions of cultures of horse isolates of *Salmonella* serovars *Coeln* and *Enteritidis*. It remains to be determined whether this correlation also applies to clinical samples, warranting further investigations.

In another study comparing culture and qPCR results, the median qPCR Ct values of culture negative but PCR positive samples was 35.39 (range 22.03–36.95) in 10 horses with gastrointestinal signs and 34.70 (range 31.98–37.00) in 7 horses without gastrointestinal signs [39]. These values were very close to the positivity threshold used in this study (namely a Ct value ≤ 37.00) and were interpreted by the authors as the possible presence of a low bacterial load in these samples.

It should, however, be emphasized that the estimation of the infectious load of a sample based on the Ct value of the qPCR is considered to be a very approximate technique. Indeed, many variables can influence this value, such as the technique and type of sample used, the time and conditions for transport and storage of the sample, the DNA extraction technique, the efficiency of the quantitative PCR test, and the method for determining the Ct values [44]. Normalization of the Ct values has sometimes been used to reduce these errors [70]. Such a correction method was not used here. However, three Salmonella strains isolated from horse samples were used to establish standard curves correlating the bacterial concentration (log CFU/mL) with the Ct value observed after DNA extraction and qPCR, according to our standard diagnostic protocol. These three strains corresponded to two different serovars (one strain of *Salmonella enterica* ssp. *enterica* serovar *Coeln* and two strains of the *Enteritidis* serovar). First, an excellent correlation was observed between the qPCR Ct value and the bacterial concentration (R^2^ values of 0.9983, 0.9973 and 0.9997, respectively). Next, although the equations differed slightly, the bacterial concentration estimated based on the Ct value was relatively similar (for instance, for a Ct value of 30, a standard deviation of 15.9% was observed between the results obtained with the three equations). It was therefore decided to average the three equations to obtain a single equation for predicting log CFU/mL from the qPCR Ct value. However, the number of strains used is very limited, and it would have been useful to check whether the prediction equation would have been comparable with other strains. So, overall, even if the qPCR Ct value appears as an interesting indicator of the actual bacterial concentration, further studies are needed to confirm this correlation on a broader range of serovars; then in a context better mimicking the clinical samples (for instance, spiking horse feces with serial dilutions of bacterial cultures); and, finally, directly on clinical samples. Despite all the aforementioned limitations, the results suggest a link between the Ct values of the qPCR test and the severity of the clinical signs in horses shedding *Salmonella*. The use of the Ct values could therefore make it possible to refine the short-term interpretation of a positive PCR result. Indeed, during the delay to obtain the culture results, strict biosecurity measures, which are restrictive and costly, could be implemented only for horses below a certain threshold, while lighter measures could be applied to horses with higher Ct values. In addition, even if this will obviously not be the only decision criteria, cases with low Ct should be placed under particularly close clinical monitoring, or even under preventive treatment for endotoxemia because they seem to present a greater risk of unfavorable clinical evolution.

This study has several other limitations. As a retrospective study, there is a risk of inaccuracies in the collection of clinical data. The determination of the SIRS score was based on the four criteria usually used in equine medicine [45], but those signs are non-specific and may not always reflect the severity of the clinical disease. Moreover, in a few cases, the circulating WBC count was not measured at the time of sampling, which may have led to underestimation of the SIRS score, as in these samples it was determined on the basis of three rather than four criteria. Another limitation of the study is that the efficacy and efficiency of the assay was not studied with samples known to be negative for *Salmonella* based on culture and standard PCR, with other qPCR assays using other genes, or using samples spiked with known quantities of one or several *Salmonella* serovars. On the basis of the Moore and Feist (2007) study results [47], the *Stn* qPCR was considered to be an assay that can be used with good reliability in the context of a diagnostic laboratory, but additional validation studies should ideally be carried out to refine the quantitative interpretation of the results. Moreover, samples positive by PCR were not systematically submitted for culture (currently considered the gold standard for diagnosis of salmonellosis [5,37]), which did not allow study of the link between the Ct of the qPCR and the culture results. Finally, even if all the qPCR tests were carried out in the same laboratory and using the same methodology, they were not carried out simultaneously, which could be a source of bias and could have influenced the Ct values obtained. To limit this effect, strictly standardized procedures were used and the number of laboratory technicians involved was limited to three people over the two year period of the study. Despite all those limitations, determination of a Ct threshold value for the results obtained in a laboratory, according to a standardized methodology, could be an interesting indicator for a finer interpretation of the results, rather than a simple negative or positive result. In the future, it could be very useful to compare the sensitivity and specificity of the *Stn* qPCR assay used in this study with other assays, including bacterial isolation, DNA quantification, and serotyping, as a diagnostic tool of *Salmonella* spp. shedding in horses.

## 5. Conclusions

In conclusion, the results of this study suggest that taking into account the Ct values in the interpretation of a qPCR result could improve the diagnosis of clinical salmonello-sis in horses. This could help in interpreting the results of a positive fecal qPCR test, especially in hospitalized patients. Indeed, clinical monitoring and biosecurity measures could be reinforced, and the prognosis could be announced as less favorable to the owner if the Ct values of the test are low. On the contrary, results with high Ct values could be less alarming, and less restrictive biosecurity measures and clinical follow-up could be applied on those cases. However, as the Ct depends on several procedures and primer-dependent factors, each laboratory should establish its own standard cut-off values.

## Figures and Tables

**Figure 1 microorganisms-11-01950-f001:**
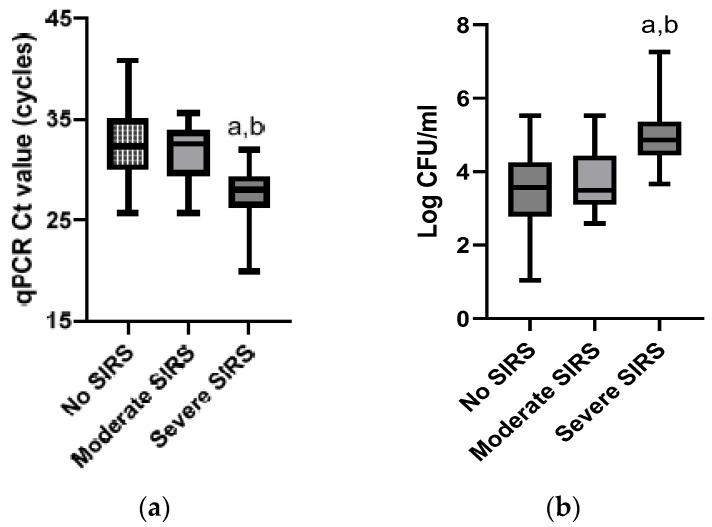
Boxplots of the cycle threshold value (Ct) of positive *Salmonella* spp. qPCR tests (**a**) and of the corresponding log CFU/mL estimated from standard calibration curves (**b**) obtained on fecal samples from horses hospitalized for gastrointestinal problems and not presenting SIRS (*n* = 72), presenting a moderate SIRS (*n* = 26), or presenting a severe SIRS (*n* = 22) at the time of sampling. a and b = significantly different from samples obtained on cases with no SIRS or on cases with a moderate SIRS, respectively, using the one-way ANOVA test (*p* < 0.05).

**Figure 2 microorganisms-11-01950-f002:**
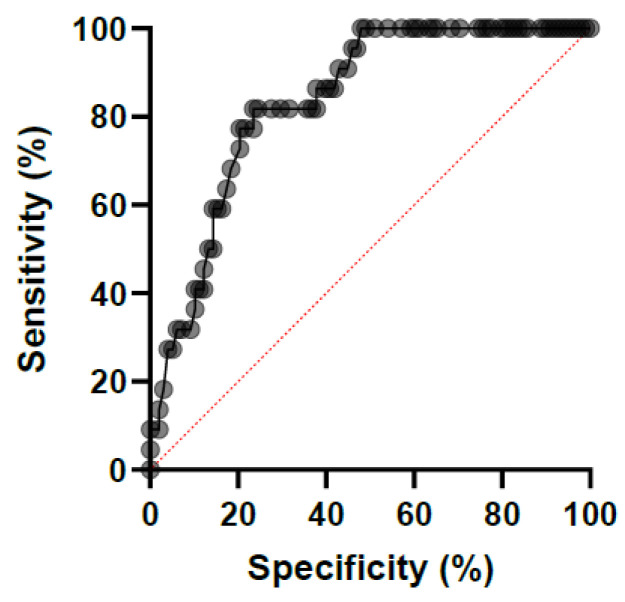
Receiver-operating characteristic curves showing the performance of the cycle threshold value (Ct) of a *Salmonella* spp. qPCR performed on fecal samples in predicting the association with a severe systemic inflammatory response syndrome (SIRS) in 82 horses hospitalized for gastro-intestinal problems.

**Figure 3 microorganisms-11-01950-f003:**
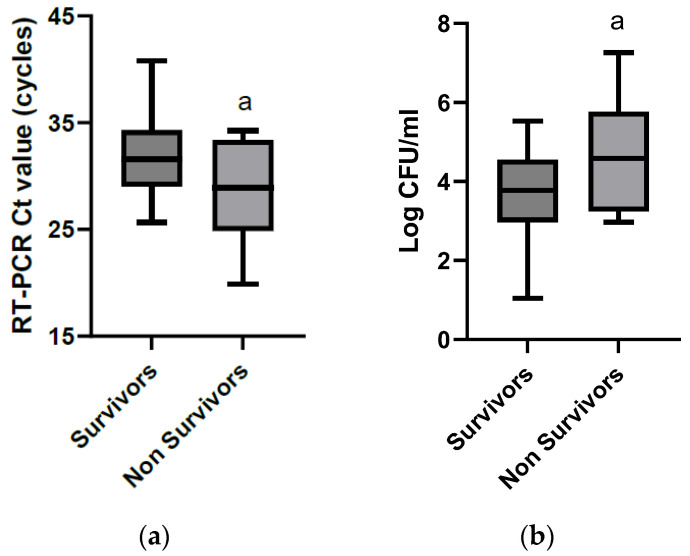
Boxplots of the cycle threshold value (Ct) of positive *Salmonella* spp. qPCR tests (**a**) and of the corresponding log CFU/mL estimated from standard calibration curves (**b**) obtained on fecal samples from horses hospitalized for gastrointestinal problems that were discharged from the hospital (survivors, *n* = 73) or that died or were euthanized during hospitalization (non-survivors, *n* = 9). a = significantly different from samples obtained on surviving cases, using the unpaired *T*-test (*p* < 0.05).

## Data Availability

Raw data are available from the corresponding author on reasonable request.

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
