# Peer review of "Relationship between the Cycle Threshold Value (Ct) of a Salmonella spp. qPCR Performed on Feces and Clinical Signs and Outcome in Horses"

_microorganisms, 2023, doi:10.3390/microorganisms11081950_

Round 1

Reviewer 1 Report

Dear authors,

The work seems to be important and practical. However, I believe there could be some improvements in the validation process. I will try to be brief in saying what would suggest for steps to consider:

Writing:

Introduction - I would suggest describing what potential serovars of Salmonella are important for this case, and any previous comparison that have been attempted using other broad range primers to set the stage for the gap of knowledge that this work fits in.

Methods

I would suggest describing with more detail the amount of DNA used in the reactions, how quantification was done, normalization of concentration, number of cycles of qPCR run. Perhaps, it would be nice to see a table with the description of horse characteristics and disease in question.

Steps in the validation process:

1. I really think that this paper would benefit of using specific serovar(s) that are associated with this disease to 1) construct grow curves from which DNA can be used in known and normalized quantity to make standard curves for the qPCR. In that way the CFU counts can be correlated directly with Ct values, because it is hard to know what the ration of Ct value to log10 of bacteria, and you can significant differences in Ct value reflecting very little change in the bacterial abundance. That does not limit the application of using Ct values to call a sample positive or negative.

2. I would then assess the efficacy of the assay with samples know to be negative for Sal based on culture and standard PCR, and other qPCR assays using other genes as listed in the paper, and in some I would spike known quantities of Sal (one serovar or multiple) to assess efficiency

3. I would most definitely use clinical samples like it was used but by providing a comparison of previous microbiological and PCR results. In a subset, you could even assess this primer set in comparison with the other two genes that are used

4. it seems that the stn gene may not be present in bongori or other Enterobacteriacea (https://pubmed.ncbi.nlm.nih.gov/8580901/), but that could be tested for comparison a bioinformatics search using blast, and a few testing with samples negative for sal by other assays, and by testing against Escherichia coli for instance

5. If in this setting you have access to samples of other animals that also have confirmed cases of Sal, you could confirm its efficacy that way, especially, by selecting cases associated with different serovars (a few)

I believe the work is important , but I also think the validation process can be more thorough.

Sincerely

Reviewer 2 Report

The manuscript by Amory et al. is original and well written, suggesting that the inclusion of the threshold value in the interpretation of fecal qPCR results could improve the diagnostic value of this test for clinical salmonellosis in horses. I support its possible further processing after clarification of some concerns as highlighted below:

- please ensure that the scientific name of the all species are written in italic (e.g. L34 Salmonella, etc.);

- please use rectangular brackets for citations according to the journal requirement;

Introduction – please improve the documentation and also mention the public health importance of Salmonella spp. for the food industry, consulting and possible citing recently published articles (e.g. https://doi.org/10.3390/foods12091756 and https://doi.org/10.3390/foods11182924)

L42: Please insert reference after the end of this sentence „... asymptomatic Salmonella shedding is not uncommon.”

L51: „an-ti-microbials” – please correct

L52: „ad-vantage” – please correct

L54: „pres-ence” – please correct, and furthere revise this typing mistake

L74: „Our hypothesis” – please avoid the using of personal mode verb formulations, it is not so characteristic for the scientific style.

L106: please specify if any positive and negative controls were used during PCR investigation

L109: „Stn” – italics and please explain/justify why this type of gene were selected as target in the materials and methods section and not at the discussion. The invA gene would be more suitable.

L120: „A p-value of 0.05 was considered significant.” – you are sure that this sentence is correct?

L123-124: the biggest concern of this manuscript is the presented results. Why no positive control was used? Why no positive control was used during PCR? Why were the positive results not confirmed by basic microbiological techniques? Why were cultures followed by serotyping not done? However, the authors correctly acknowledged these limitations. 

invL126-127: „clinical signs suggestive of salmonellosis during hospitalization” – please mention this signs (e.g. ...)

L126: when you express overall prevalence values, please indicate in brackets the value of 95% confidence interval

L161: please also include within the discussion section the possible existence of other diseases with 

L239: please discuss the possible occurrence of false positive and negative results

L280: please indicate further directions in the approached research area

L303: the reference list is not in agreement with the journal requirement. Please carefully revise this!!!

Round 2

Reviewer 1 Report

Dear authors,

I understand that the qPCR is being used for diagnostics, but without a conversion of Ct to CFU, I believe the statistical analysis can be difficult to interpret by not knowing how much change there was (predicted at least from standard curves) in bacterial counts. If one is using qPCR for positive and negatives, I can understand and appreciated that, but for quantification a standard curve would be necessary. There can be marginal significant differences in Ct values between samples that perhaps would not be strong in the predicted bacterial concentration. I believe the work is important but a standard curve with major clinical serovars would be necessary.

Thank you

Author Response

see here enclosed comments

Reviewer 2 Report

Much improved!

Author Response

We thank you for this positive comment.

Round 3

Reviewer 1 Report

Dear authors, 

Thank you for your response and extra work on the standard curves. I believe that complementary to what has been done so far, I would suggest that all three figures could be improved the following way: by adding 4 plots to each, by converting the current qPCR Ct values to the predictive CFU counts according to each specific standard curve (each of the three isolates), or an average of all standard curves across all serovars (mean of all 3 isolates). That way each figure will have an extra 4 plots and regardless of the limitations, it will at least demonstrate the predictability of bacterial load in a more translatable way than Ct values only. Please, I would also include the appropriate statistics as well. 

Sincerely

Author Response

Please find our answer to reviewer 1 here enclosed
